# EGF Conjugation Improves Safety and Uptake Efficacy of Titanium Dioxide Nanoparticles

**DOI:** 10.3390/molecules25194467

**Published:** 2020-09-29

**Authors:** Basma Salama, Chia-Jung Chang, Koki Kanehira, El-Said El-Sherbini, Gehad El-Sayed, Mohamed El-Adl, Akiyoshi Taniguchi

**Affiliations:** 1Research Center for Functional Materials, National Institute for Materials Science (NIMS), 1-1 Namiki, Tsukuba, Ibaraki 305-0044, Japan; dr.basma_bio@yahoo.com (B.S.); CHANG.Chiajung@nims.go.jp (C.-J.C.); koki.kanehira@jp.toto.com (K.K.); 2Department of Biochemistry and Chemistry of Nutrition, Faculty of Veterinary Medicine, Mansoura University, 60 El Gomhouria St., Mansoura, Dakahlia Governorate 35516, Egypt; sshrbini@mans.edu.eg (E.-S.E.-S.); grelsayed@yahoo.com (G.E.-S.); drmohamedalymaher@hotmail.com (M.E.-A.); 3Inorganic Materials Research Section, TOTO Ltd. Research Institute, Honson 2-8-1, Chigasaki, Kanagawa 253-8577, Japan

**Keywords:** titanium dioxide nanoparticles, EGF, EGFR, cellular uptake, cell proliferation

## Abstract

Titanium dioxide nanoparticles (TiO_2_ NPs) have a strong potential for cancer therapeutic and bioimaging applications such as photodynamic therapy (PDT) and photodynamic diagnosis (PDD). Our previous results have shown that TiO_2_ NPs have a low cellular uptake and can induce cell proliferation. This suggests that TiO_2_ NPs could increase the risk of tumor overgrowth while being used for PDD and PDT. To solve this problem, we constructed epidermal growth factor-ligated polyethylene glycol-coated TiO_2_ NPs (EGF-TiO_2_ PEG NPs). In this work, we studied the effect of EGF conjugation on the cellular uptake of TiO_2_ PEG NPs. Then, we investigated the effect of both non-conjugated and EGF-TiO_2_ PEG NPs on the A431 epidermal cancer cell line, proliferation and growth via the investigation of EGFR localization and expression. Our results indicated that TiO_2_ PEG NPs induced EGFRs aggregation on the A431 cells surface and induced cell proliferation. In addition, EGF-TiO_2_ PEG NPs induced the internalization of EGFRs inside of cells with increased cellular NPs uptake and decreased cellular proliferation compared to TiO_2_ PEG NPs-treated cells. These findings suggest that EGF conjugation can increase the efficacy of TiO_2_ PEG NPs for biomedical applications such as PDD and PDT with decreased risk of tumor overgrowth.

## 1. Introduction

Titanium dioxide nanoparticles (TiO_2_ NPs) have a high potential to be applied in nanomedicine based on their unique properties, including biocompatibility, photocatalytic activity, and light scattering properties [1,2]. TiO_2_ NPs have a potential in cancer therapy and bioimaging applications such as photodynamic therapy (PDT) and photodynamic diagnosis (PDD) [3,4]. In PDT, TiO_2_ NPs are used as a photosensitizer. TiO_2_ NPs generate reactive oxygen species (ROS) and induce cell death when activated by irradiation at an appropriate wavelength [5]. PDD is a very effective technique for cancer localization and intraoperative tumor visualization, and TiO_2_ NPs have been used to enhance its fluorescence signals [6,7]. Recent work has found that TiO_2_ NPs can decrease photobleaching and enhance and prolong fluorescent PDD agents, resulting in improved tumor visualization [8]. Moreover, it has been recommended to combine PDT and PDD for the simultaneous treatment and detection of cancers [9,10,11]. These basic studies show that TiO_2_ NPs are excellent agents for PDD and PDT. For clinical application of TiO_2_ NPs, it is important to investigate their safety. Therefore, a basic in vitro study of the interactions between TiO_2_ NPs and cells is necessary prior to clinical application.

Previously, we investigated the interaction between TiO_2_ NPs and cancer cell lines to clarify the safety and cellular uptake efficacy of TiO_2_ NPs [12,13,14]. Our results showed that 100 nm TiO_2_ NPs had a low cellular uptake by cancer cell lines. Moreover, TiO_2_ NPs induced HepG2 and A431 cell proliferation [14]. TiO_2_ NPs can associate with hepatocyte growth factor receptors (HGFRs), leading to receptor aggregation, facilitating the recruitment of more signal transduction molecules, and HepG2 cell proliferation and growth [13]. These results suggest that TiO_2_ NPs can increase the risk of tumor overgrowth while being used for PDD and PDT. To solve this problem, we constructed epidermal growth factor-ligated polyethylene glycol-coated TiO_2_ NPs (EGF-TiO_2_ PEG NPs) aiming to increase NP cellular uptake and decrease their cell proliferative effect. The target of EGF-TiO_2_ PEG NPs is the epidermal growth factor receptor (EGFR), which is abnormally overexpressed in many cancer types and plays a key role in promoting cell proliferation and opposing apoptosis [15]. Therefore, EGFR is one of the most important candidates for targeted cancer therapy and diagnosis by ligand-targeted NPs [16]. We assumed that EGF-TiO_2_ PEG NPs could bind with EGFR due to the presence of the specific ligand, EGF, on the NP surface. The binding of EGF-TiO_2_ PEG NPs with EGFR should induce receptor-mediated endocytosis of the receptor-NP complex. This receptor-mediated endocytosis could lead to an increase of cellular NP uptake and a decrease of EGFR cell surface expression, resulting in a reduction of cell proliferation signals.

In this work, we investigated the effect of EGF conjugation on the cellular uptake level of TiO_2_ PEG NPs. Then, we investigated the effect of both non-conjugated and EGF-TiO_2_ PEG NPs on cancer cell line proliferation and growth via the investigation of EGFR localization and expression. For this purpose, we used A431 cells, which are derived from epithelial cell carcinoma. The A431 cell line is as a well-known model for EGFR-overexpressing cancer [17]. Our results indicated that TiO_2_ PEG NPs induced EGFR aggregation on the A431 cells surface and induced cell proliferation and growth. In addition, EGF-TiO_2_ PEG NPs induced the internalization of EGFRs inside of cells with an increased cellular NPs uptake level and decreased cellular proliferation compared to TiO_2_ PEG NPs-treated cells. These findings suggest that EGF conjugation increases the efficacy of TiO_2_ PEG NPs for biomedical applications such as PDT and PDT with decreased risk of tumor overgrowth.

## 2. Results

### 2.1. EGF Conjugation Increased TiO_2_ PEG NPs Uptake by A431 Cells

We constructed EGF-conjugated TiO_2_ PEG NPs, as illustrated in Figure 1, so as to improve NPs uptake by EGFR-expressing cells. The average hydrodynamic particle sizes of TiO_2_-PEG NPs and EGF-conjugated TiO_2_ PEG NPs in water were 120 and 128 nm, respectively.

We evaluated the effect of EGF conjugation on the cellular uptake level of TiO_2_ PEG NPs. A431 cells were treated with 10 and 100 µg/mL of TiO_2_ PEG NPs and EGF-TiO_2_ PEG NPs, and cellular uptake was evaluated based on the change in cell side-scattering by flow cytometry. As shown in Figure 2, A431 cells treated with 10 µg/mL EGF-TiO_2_ PEG NPs showed an approximately two-times higher uptake ratio compared with the 10 µg/mL TiO_2_ PEG NPs treatment. A431 cells treated with 100 µg/mL of EGF-TiO_2_ PEG NPs showed an approximately 1.5-times higher uptake ratio compared with the 100 µg/mL of TiO_2_ PEG NPs treatment. These results indicated that EGF conjugation enhanced the cellular uptake of TiO_2_ PEG NPs by A431 cells.

### 2.2. EGF Conjugation Decreased the Proliferative Effect of TiO_2_ PEG NPs on A431 Cells

Previously, our research showed that a low concentration of TiO_2_ PEG NPs induces HepG2 cells to proliferate through the HGFR receptor aggregation [13], and we have also found that TiO_2_ PEG NPs can induce A431 cell proliferation [14]. These findings suggested that TiO_2_ PEG NPs could increase the risk of tumor overgrowth. Therefore, we conjugated EGF with TiO_2_ PEG NPs to modulate the interaction between TiO_2_ PEG NPs and cells and consequently modulate cancer cell proliferation.

A431 cells were treated with 10 μg/mL of TiO_2_ PEG NPs and EGF-TiO_2_ PEG NPs, and cell viability assays were performed to evaluate cell proliferation. As shown in Figure 3A, TiO_2_ PEG NPs increased A431 cell viability up to 160% compared to control nontreated cells. In contrast, EGF-TiO_2_ PEG NPs-treated cells showed a significant decrease in cell viability compared to TiO_2_ PEG NPs-treated cells. Our results suggested that EGF conjugation decreased the proliferative effect of TiO_2_ PEG NPs, and subsequently increasing their safety. To verify the results of the cell viability assays, A431 cells were stained by trypan blue and counted by a hemocytometer after exposure to 10 μg/mL of TiO_2_ PEG NPs and EGF-TiO_2_ PEG NPs. As shown in Figure 3B, TiO_2_ PEG NPs significantly increased A431 cell numbers compared to control nontreated cells. In contrast, EGF-TiO_2_ PEG NPs-treated cell numbers were significantly decreased compared to TiO_2_ PEG NPs-treated cells.

The result suggested that EGF conjugation to TiO_2_ PEG NPs decreased cell numbers by cell growth decrease. However, another possibility is cell numbers decreased by cell death. To determine whether EGF conjugation decreased cell numbers by cell growth decrease, 5-ethynyl-2′-deoxyuridine (EdU) proliferation assays were performed by counting the percentage of A431 cells with incorporated EdU, which indicates newly synthesized DNA (shown in green). As shown in Figure 4, EdU positive cells were decreased by EGF conjugation. The result indicated that EGF conjugation to TiO_2_ PEG NPs significantly decreased cell numbers by DNA synthesis decrease, suggesting cell growth decrease. These results verified that EGF conjugation to TiO_2_ PEG NPs decreased the risk of cancer growth and subsequently increased the safety of TiO_2_ PEG NPs for biomedical applications.

### 2.3. Expression and Localization of EGFR by Immunofluorescence Staining

To understand the molecular mechanism underlying the effect of EGF conjugation on the TiO_2_ PEG NPs cellular uptake and cell proliferation, we investigated the effect of non-conjugated and EGF-conjugated TiO_2_ PEG NPs on EGFR localization and expression by immunofluorescence staining. EGFR is a cell membrane receptor that plays a key role in cell proliferation. A431 cells, which are considered model EGFR-overexpressing cancer cells, were treated with TiO_2_ PEG NPs and EGF-TiO_2_ PEG NPs, then the localization of EGFR was visualized by immunofluorescence staining. Confocal microscopy images showed that EGFR was regularly distributed on the surface of nontreated A431 cells (Figure 5, upper panels). For TiO_2_ PEG NPs-treated cells, EGFR aggregated on the cell surface (Figure 5, middle panels). In EGF-TiO_2_ PEG NPs-treated cells, EGFR localization changed to the inside of the cells (Figure 5, lower panels). The same results were obtained in different focus pictures (data not shown). Our finding indicated that TiO_2_ PEG NPs induced cell proliferation via EGFRs aggregation. In contrast, EGF-TiO_2_ PEG NPs interacted with EGFR and induced the receptors to internalize inside the cells. The EGFR cell surface expression level was subsequently decreased, leading to a decrease in signals for cell proliferation.

## 3. Discussion

The biocompatibility, photocatalytic activity, and light scattering properties of TiO_2_ PEG NPs are the reasons behind their interest for cancer therapy and diagnosis techniques such as PDT and PDD. Therefore, it is very beneficial to investigate the safety of TiO_2_ PEG NPs as used in nanomedicine. Our previous research concluded that TiO_2_ PEG NPs have a relatively low cellular uptake ratio [12,13,14]. In addition, TiO_2_ PEG NPs induce HepG2 and A431 cell proliferation [14]. TiO_2_ PEG NPs can interact with HGFRs on the surface of HepG2 cells, leading to receptor aggregation and HepG2 cell proliferation [13]. Our findings have suggested that TiO_2_ PEG NPs can increase the risk of tumor growth when used for cancer bioimaging and cancer therapy. For this reason, we constructed EGF-conjugated TiO_2_ PEG NPs to improve their cellular uptake level and decrease their proliferative effect. The results of this paper indicated that EGF-conjugation improved TiO_2_ PEG NPs cellular uptake with decreased cell proliferation.

EGFR, a transmembrane receptor that is overexpressed in many cancer types, plays a key role in cellular proliferation [18]. Typically, EGFRs are expressed in the form of inactive monomers, but in the presence of a specific ligand, EGFRs aggregate to form active dimers that induce signal transduction for cell proliferation [19]. We assumed that EGF conjugation on TiO_2_ PEG NPs surface would induce the binding of NPs with EGFR. The binding of EGF-TiO_2_ PEG NPs with EGFR could then induce receptor-mediated endocytosis, leading to increased NPs cellular uptake, decreased localization of EGFR on the cell surface, and decreased signaling for cell proliferation. In this time, we did not control the orientation of EGF on TiO_2_ PEG NPs surface. EGF orientated the right way on NPs surface would be improved EGF-TiO_2_ PEG NPs functions.

In this work, we investigated the effect of EGF conjugation on cellular uptake as well as the cell proliferative effect of TiO_2_ PEG NPs. We used A431 cells, an EGFR-overexpressing cancer cell line, to investigate the change in TiO_2_ PEG NP cellular uptake after EGF conjugation. A431 cells were a useful cell line for investigation of EGF effects [20]. EGF stimulates the growth of A431 cells at a low concentration, but inhibits their proliferation at a higher concentration [21]. We found that EGF-TiO_2_ PEG NPs showed a higher uptake level compared to TiO_2_ PEG NPs. Moreover, we found that EGF-TiO_2_ PEG NPs-treated A431 cells showed decreased cell proliferation and growth compared to TiO_2_ PEG NPs-treated cells. Our findings suggest that EGF conjugation increases the uptake level and decreases the cell proliferative effect of TiO_2_ PEG NPs by EGFR-overexpressing cancer cells. For the next step, we should check that EGF conjugated particles have the same therapeutic capacities as unconjugated counterparts.

We assumed that the reason behind the increased uptake of EGF-TiO_2_ PEG NPs with decreased cell proliferation is their binding to EGFR. Therefore, we investigated the localization of EGFR in A431 cells after exposure to non-conjugated and EGF-conjugated TiO_2_ PEG NPs. A431 cells treated with TiO_2_ PEG NPs showed that EGFR aggregated on its surface. In contrast, A431 cells treated with EGF-TiO_2_ PEG NPs showed localization of EGFR inside the cell cytoplasm. Fluorescence labeled EGF-conjugated TiO_2_ PEG NPs would show more sure results. These results prompted us to propose the putative molecular mechanism shown in Figure 6. For nontreated A431 cells, EGF in the culture medium induced EGFR dimerization, leading to a weak signal transduction for cell proliferation. Cells exposed to TiO_2_ PEG NPs induce EGFR aggregation on the cell surface, facilitating the recruitment of more signal transduction molecules and leading to increased signals for cell proliferation. However, EGF-TiO_2_ PEG NPs attach to EGFR and form a complex due to the presence of the specific ligand, EGF, attached to the NPs surface. Then, the receptor-NP complex could internalize inside the cytoplasm by receptor-mediated endocytosis, resulting in increased NPs uptake with decreased expression of EGFR on the cell surface, and subsequently, decreased signal transduction for cell proliferation. Ligand-independent activation of EGFRs could also happen in A431 cells. This might be also decreased by EGF-TiO_2_ PEG NPs. We have also found that polystyrene nanoparticles with EGF increased the cellular uptake of A431 cells by Clathrin-Mediated Endocytosis [22]. For the next step, the EGFR down-stream signal transduction would be checked and also the EGF antagonist would help understand the more detailed molecular mechanism.

## 4. Materials and Methods

### 4.1. Cell Line and Cell Culture

The A431 cell line, which is derived from epithelial cell carcinoma, was cultured at 37 °C and 5% CO_2_ in High Glucose Dulbecco’s modified Eagle medium (DMEM, high glucose, Nacalai Tesque, Kyoto, Japan) supplemented by 10% (*v*/*v*) heated fetal bovine serum (Biowest, Riverside, MO, USA), 100 μg/mL penicillin, and 10 μg/mL streptomycin (Nacalai Tesque). The NPs exposures were conducted in DMEM with 10% FBS. Cells were sub-cultured every two days when they reached 70–80% confluency.

### 4.2. Preparation of TiO_2_ PEG NPs

In this work, spherical, uniform 100 nm of polyethylene glycol-modified TiO_2_ nanoparticles (TiO_2_ PEG NPs) were used, according to a previous report [12]. This size of NPs showed low cytotoxicity [12]. TiO_2_ NPs were supplied by Fuji Chemical Co., Ltd. (Osaka, Japan). The evaluation of the particle size in water was performed by dynamic light scattering (Zetasizer Nano ZS, Malvern, Worcestershire, UK).

### 4.3. Preparation of EGF-TiO_2_ PEG NPs

The construction of EGF-TiO_2_ PEG NPs is shown in Figure 1. Recombinant human epidermal growth factor at 0.2 mg/mL (EGF, Thermo Fisher Scientific, Waltham, MA, USA) was mixed with 0.5% (*w*/*v*) TiO_2_ PEG NPs in a 20 mM HEPES buffer solution (pH 7.4) at 4 °C overnight. The mixture was centrifuged at 14,000× *g* for 30 min and reconstituted with endotoxin-free sterilized water. This substitution process was repeated three times in order to remove the non-absorbed EGF, then the EGF-conjugated TiO_2_ PEG NPs were collected and sonicated for 15 min. The EGF amount of the supernatants of the EGF-TiO_2_ PEG NPs after centrifugation was measured to determine the conjugation efficiency by using a protein quantification assay kit (Macherey-Nagel GmbH & Co. KG, Düren, Germany). The conjugation efficacy was more than 90% of the initial amount of EGF. The evaluation of average hydrodynamic particle size in water was carried out by dynamic light scattering. EGF would be attached on the TiO_2_ PEG NPs surface by physical adsorption.

### 4.4. Evaluation of Cellular Uptake by Flow Cytometry

Cellular uptake of TiO_2_ PEG NPs and EGF-TiO_2_ PEG NPs by A431 cells was assessed by changes in light scattering using the flow cytometric light scatter analysis [23]. Briefly, 1 × 10^6^ cells/well were seeded in 24-well plates and incubated at 37 °C and 5% CO_2_ for 24 h, then the cells were exposed to TiO_2_ PEG NPs and EGF-TiO_2_ PEG NPs in two concentrations (10 and 100 μg/mL medium) for 24 h. Then, cells were washed twice, collected by trypsinization, washed three times with PBS, dispersed in 1 mL of 6% heated fetal bovine serum in a phosphate buffer saline (HFBS/PBS) solution, and stored on ice to be analyzed within 1 h. Immediately prior to the analysis, the cells were passed through a nylon mesh (Cell Strainer Snap Cap, Falco, NY, USA), then cellular internal granularity was assessed using side-scattered light and cell size was assessed using forward-scattered light using a SP6800 spectral analyzer (Sony Biotechnology, Tokyo, Japan). The percentage of cells incorporated with nanoparticles was calculated based on changes in the gated areas compared with a control untreated population.

### 4.5. Cell Viability/Cytotoxicity Assay

The effect of TiO_2_ PEG NPs and EGF-TiO_2_ PEG NPs on the A431 cell viability was investigated using a LIVE/DEAD^®^ Viability/Cytotoxicity kit (Invitrogen, Ltd., Cambridge, UK) according to the manufacturer’s instructions using a fluorescence microplate protocol. Briefly, 1 × 10^4^ cells/well were seeded in a Costar 96-well plate and incubated at 37 °C and 5% CO_2_. After 24 h, the cells were exposed to 10 μg/mL of TiO_2_ PEG NPs and EGF-TiO_2_ PEG NPs for 24 h. The cells were then stained with 1 μM calcein AM to stain live cells and 2 μM ethidium homodimer-1 (EthD-1) to stain dead cells, then the fluorescence intensity was measured using a Spark^TM^ 10 M multimode microplate reader (Tecan Ltd., Männedorf, Switzerland).

### 4.6. Cell Counting by Trypan Blue

To avoid the interference of TiO_2_ PEG NPs with the cell viability assay by light absorption, light scattering, or fluorescence, cells were stained and counted using a disposable hemocytometer (Funakoshi, Tokyo, Japan). A431 cells were seeded at 1 × 10^5^ cells/well in 24-well plates and incubated at 37 °C and 5% CO_2._ After 24 h, cells were treated with 10 μg/mL of TiO_2_ PEG NPs and EGF-TiO_2_ PEG NPs for 24 h. Then, cells were collected by trypsinization and stained by trypan blue, and live cells were counted under a light microscope using a hemocytometer.

### 4.7. New DNA Synthesis Detecting

To monitor DNA synthesis in proliferating cells, the 5-ethynyl-2′-deoxyuridine (EdU) assay was performed as previously described [24]. Briefly, A431 cells were seeded in 4-compartment cell view cell culture dishes (Greiner Bio-One, Inc., Monroe, NC, USA) at a density of 3 × 10^4^ cells/compartment at 37 °C and 5% CO_2_ for 24 h, then the cells were exposed to TiO_2_ PEG NPs and EGF-TiO_2_ PEG NPs (10 μg/mL in culture medium). After 24 h, the cells were incubated with a 10 μM EdU solution for 2 h at 37 °C. Subsequently, the cells were fixed with 4% formaldehyde for 15 min. After rinsing with 3% BSA in phosphate buffered saline (PBS, pH 7.4, Sigma-Aldrich, St. Louis, MO, USA), cells were permeated with 0.5% Triton X-100 in PBS, incubated with iFluor 488 azide, and stained with 300 nM 4′,6-diamidino-2-phenylindole (DAPI, Abcam) for 20 min. All images were acquired with a Zeiss LSM 510 META confocal microscope system (LSM510 META, Carl Zeiss Inc., Jena, Germany). At least 400 nuclei were counted per experiment.

### 4.8. Immunofluorescence Staining of EGFRs

A431 cells were seeded in 4-compartment cell view cell culture dishes (Greiner Bio-One, Inc., Monroe, NC, USA) at a density of 2.5 × 10^4^ cells/compartment at 37 °C and 5% CO_2_ for 24 h. Next, the cells were exposed to 10 μg/mL of TiO_2_ PEG NPs and EGF-TiO_2_ PEG NPs. After 24 h, cells were washed with PBS and fixed with 4% paraformaldehyde for 10 min, then permeabilized with 0.1% Triton X-100. Subsequently, the cells were blocked with 1% bovine serum albumin/10% normal goat serum/0.3 M glycine in 0.1% Tween-PBS for 1 h at room temperature, then incubated with anti-EGFR antibody (1/500 dilution, Thermo Fisher Scientific, Runcorn, UK) at 4 °C overnight. The cells were washed three times (10 min each) with PBS and incubated with goat anti-mouse IgG H&L (1/500 dilution, Abcam) in the dark for 1 h at room temperature, followed by washing three times (10 min each) with PBS. Nuclear DNA was labeled with DAPI (Thermo Fisher Scientific, Waltham, MA, USA). Images were taken using a confocal laser scanning microscope.

### 4.9. Statistical Analysis

All data were assessed for statistical significance using the Student’s *t*-test and one-way Analysis of Variance (ANOVA). All values are presented as means ± SD with three or more independent replicates (*n* ≥ 3), ** *p* ≤ 0.01 and *** *p* ≤ 0.001, which are indicated in the figure legends.

## 5. Conclusions

In this paper, we showed that EGF conjugation increased the safety and cellular uptake of TiO_2_ PEG NPs when used on EGFR-expressing cancer cells via their interaction with EGFR. Our results suggest that EGF-TiO_2_ PEG NPs can be effectively used in cancer bioimaging methods such as PDD and for cancer therapies such as PDT with a decreased risk of cancer overgrowth.

## Figures and Tables

**Figure 1 molecules-25-04467-f001:**
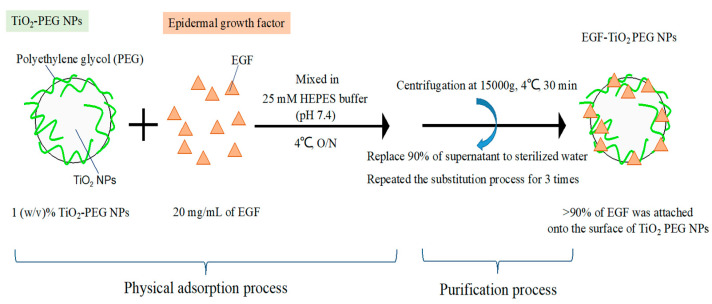
Construction of epidermal growth factor-ligated polyethylene glycol-coated titanium dioxide nanoparticles (EGF-TiO_2_ PEG NPs). EGF was mixed with TiO_2_ PEG NPs in a 20 mM HEPES buffer solution (pH 7.4) at 4 °C overnight. The mixture was centrifuged and reconstituted with free sterilized water. This substitution process was repeated three times, then EGF-conjugated TiO_2_ PEG NPs were collected. The conjugation efficacy was more than 90% of the initial amount of EGF.

**Figure 2 molecules-25-04467-f002:**
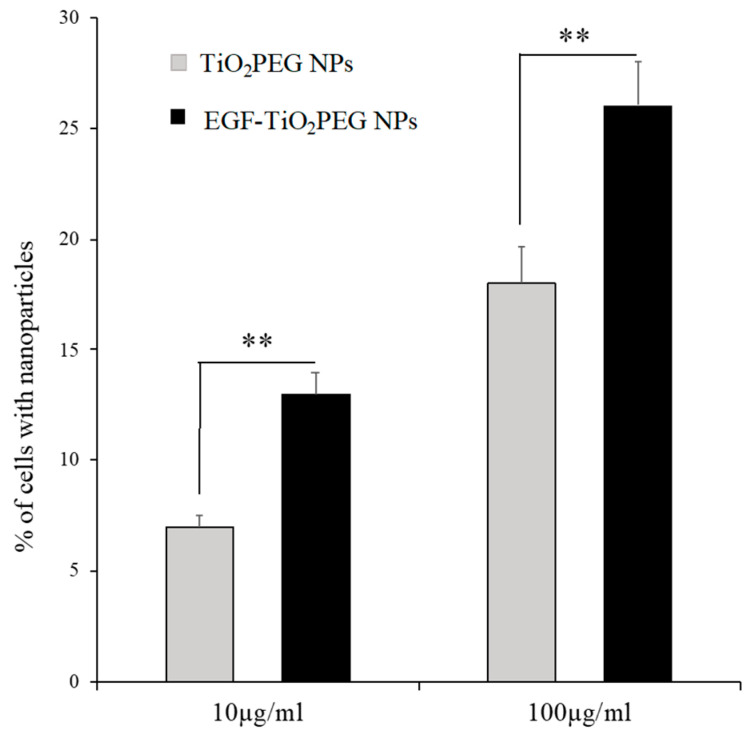
The effect of EGF conjugation on TiO_2_ PEG NPs uptake by A431 cells. The percentage of A431 cells incorporated with NPs was assessed by flow cytometry after exposure to TiO_2_ PEG NPs (gray bars) and EGF-TiO_2_ PEG NPs (black bars) at a concentration of 10 μg/mL (left bars) and 100 μg/mL (right bars) for 24 h. All values are presented as means ± SD (*n* ≥ 3). Data were analyzed using the Student’s *t*-test; ** *p* ≤ 0.01.

**Figure 3 molecules-25-04467-f003:**
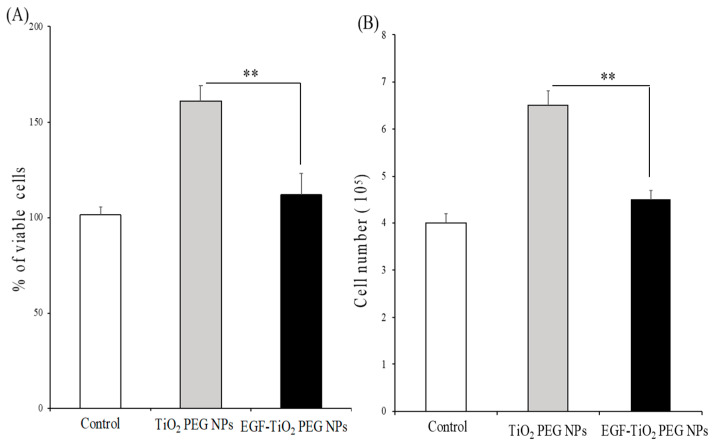
Effect of non-conjugated and EGF-conjugated TiO_2_ PEG NPs on A431 cell viability (**A**) and growth (**B**). A431 cells were treated by TiO_2_ PEG NPs (gray bar) and EGF-TiO_2_ PEG NPs (black bar) at a concentration of 10 μg/mL for 24 h. The effect of NPs on cell viability and growth was calculated compared to control untreated cells (white bar). All values are presented as means ± SD (*n* ≥ 3). Data were analyzed using the Student’s *t*-test; ** *p* ≤ 0.01.

**Figure 4 molecules-25-04467-f004:**
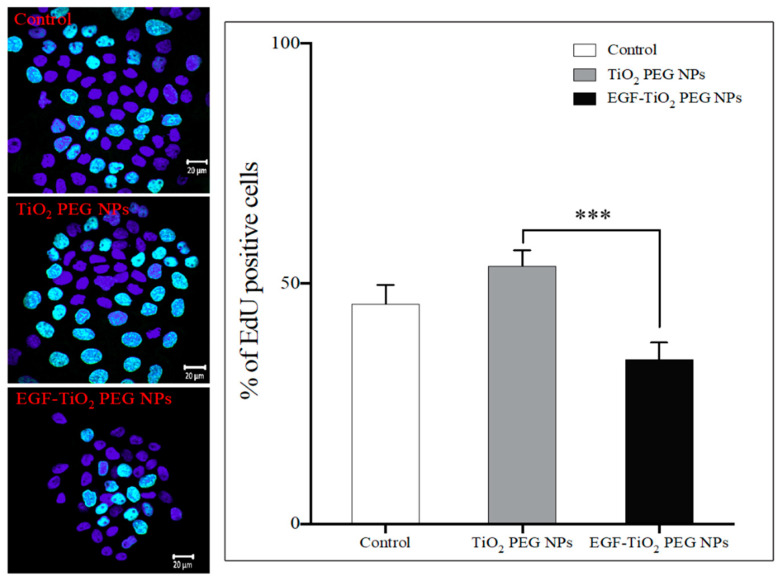
Effect of non-conjugated and EGF-conjugated TiO_2_ PEG NPs on A431 DNA synthesis. Fluorescence images (left photos) show EdU incorporation into the nuclei of cells. EdU-positive cells are green and nuclei stained with 4’,6-diamidino-2-phenylindole (DAPI) are blue. Right graph shows the percentage of EdU-positive cells. Data are reported as means ± SEM for three independent experiments. Cell numbers were compared to control untreated cells (white bar). All values are presented as means ± SD (*n* ≥ 6). Data were analyzed using the analysis of variance (ANOVA); *** *p* ≤ 0.001.

**Figure 5 molecules-25-04467-f005:**
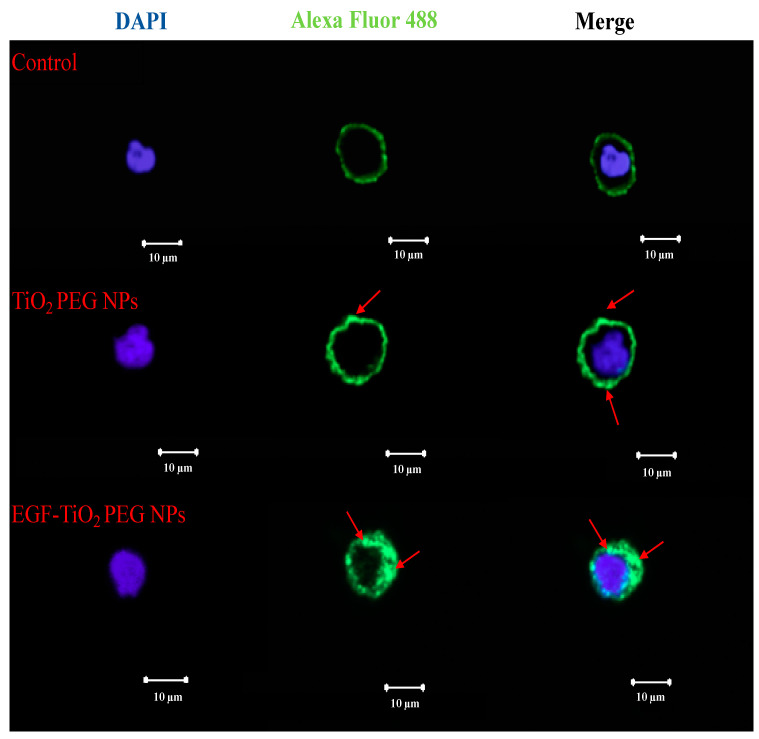
Effect of non-conjugated and EGF-conjugated TiO_2_ PEG NPs on epidermal growth factor receptor (EGFR) localization in A431 cells. A431 cells were incubated without (upper panel) or with 10 μg/mL of TiO_2_ PEG NPs (middle panel) and EGF-TiO_2_ PEG NPs (lower panel) for 24 h followed by immunofluorescence staining with anti-EGFR antibody (green). The nuclei were stained with 4′,6-diamidino-2-phenylindole (DAPI) (blue), and fluorescence images were taken with a confocal microscope. Scale bar = 10 μm.

**Figure 6 molecules-25-04467-f006:**
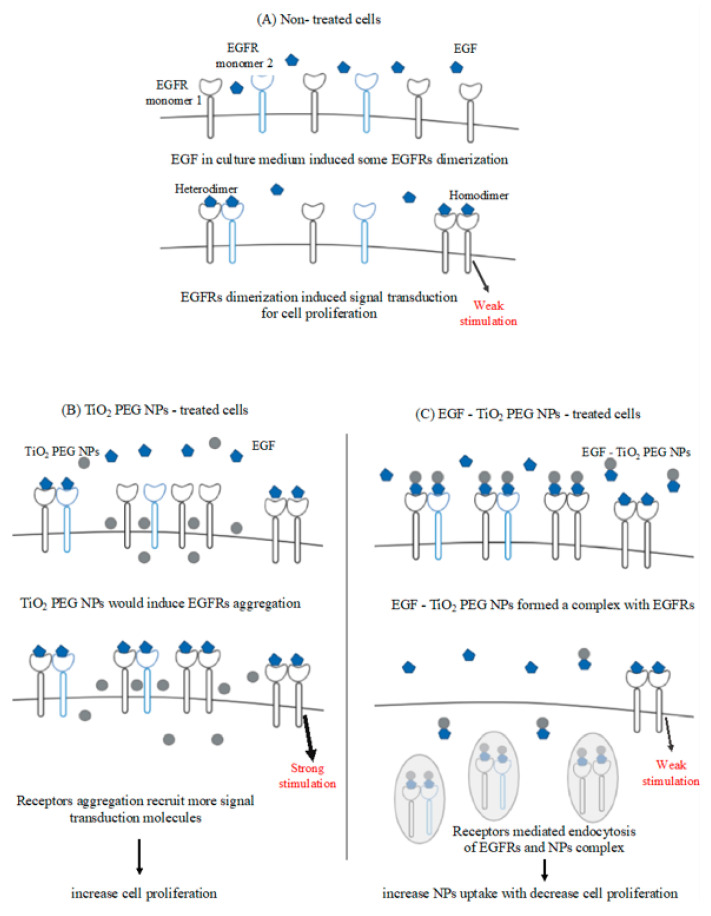
Proposed putative molecular mechanism for the effect of EGF conjugation on TiO_2_ PEG NPs uptake levels and the cell proliferation effect via interaction with EGFRs. (**A**) Represents non-treated A431 cells, (**B**) represents TiO_2_ PEG NPs-treated A431 cells, (**C**) represents EGF-TiO_2_ PEG NPs-treated cells.

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
