# Peer review of "EGF Conjugation Improves Safety and Uptake Efficacy of Titanium Dioxide Nanoparticles"

_molecules, 2020, doi:10.3390/molecules25194467_

Round 1

Reviewer 1 Report

The manuscript by Salama et al. describes the conjugation of titanium dioxide nanoparticles with EGF and the effects on A431 cell, investigating the NPs uptake, EGF receptor internalization and cell proliferation.

In my opinion the work should be improved in order to specify the molecular mechanisms involved

  • The assumption that the internalization of EGFR could be driven by the presence of EGF on NPs surface is not confirmed by the experiments conducted. A pharmacological approach is needed to demonstrate EGFR internalization through interaction with EGF conjugated NPs. For example, using an EGF antagonist, it is possible to determine if the increased NPs internalization is really caused by the receptor mediated endocytosis of EGFR and NP complexes.
  • in addition, what about the analysis of the conformation of EGF on NPs surface? To interact with its cognate receptor a protein must be oriented in the right way
  • the NPs exposures were conducted in complete cell culture medium (with FBS) or not?

Author Response

Dear Sir,

Thank you for giving us the opportunity to submit a revised draft of our manuscript to Molecules. We appreciate the time and effort that you and the reviewers have dedicated to providing your valuable feedback on our manuscript. We are grateful to the reviewers for their insightful comments on our manuscript. We have been able to incorporate changes to reflect most of the suggestions provided by the reviewers. We have highlighted the changes within the manuscript.

Reviewer 1

  • The assumption that the internalization of EGFR could be driven by the presence of EGF on NPs surface is not confirmed by the experiments conducted. A pharmacological approach is needed to demonstrate EGFR internalization through interaction with EGF conjugated NPs. For example, using an EGF antagonist, it is possible to determine if the increased NPs internalization is really caused by the receptor mediated endocytosis of EGFR and NP complexes.

Answer: This is very good idea. We have tried to block EGFR using anti-EGFR antibody. However, cell size became increasing by antibody binding, that is why, it could not detect NPs uptake by FACS. We have discussed this point in discussion section. (page 7, lanes 207 to 208)

  • In addition, what about the analysis of the conformation of EGF on NPs surface? To interact with its cognate receptor a protein must be oriented in the right way.

Answer: We did not control EGF orientation. If we control EGF orientation on NPs surface, the effect will be improved. We have added this point in discussion section. (page 7, lanes 176 to 178)

  • The NPs exposures were conducted in complete cell culture medium (with FBS) or not?

Answer: We have used with FBS condition. We have added this information in materials and methods section. (page 8, lanes 217 to 218)

Sincerely,

Akiyoshi Taniguchi PhD

Research Center for Functional Materials,

National Institute for Materials Science

1-1, Namiki, Tsukuba, Ibaraki 305-0044 Japan

Reviewer 2 Report

I the article is lack of information about the specific labelling of tested nanosystems for tracking their localization and uptake mechanism in the treated cells. - it should be provided

In the abstract, I suggest to use the full name of the cell culture line which was used.

What is the reason of increased proliferation after TiO2 treatment? - might author provide some theory ?

The obtained results should be discussed with a previously published study to indicate the significance of the proposed study and novelyty

Author Response

Dear Sir,

Thank you for giving us the opportunity to submit a revised draft of our manuscript to Molecules. We appreciate the time and effort that you and the reviewers have dedicated to providing your valuable feedback on our manuscript. We are grateful to the reviewers for their insightful comments on our manuscript. We have been able to incorporate changes to reflect most of the suggestions provided by the reviewers. We have highlighted the changes within the manuscript.

Reviewer 2

  • The article is lack of information about the specific labelling of tested nanosystems for tracking their localization and uptake mechanism in the treated cells. - it should be provided.

Answer: Thank you for your good suggestion. Sometime labeling of NPs could change the uptake ratio and localization compared to non-labeling NPs due to changing surface character. And also it was very difficult to fluorescence label to EGF-PEG-TiO2NPs. We have added this point in discussion section. (page 7, lanes 194 to 195)

  • In the abstract, I suggest to use the full name of the cell culture line which was used.

Answer: According to reviewer’s comment, we have added “A431 epidermal cancer cell line” in abstract. (page 1, lane 21)

  • What is the reason of increased proliferation after TiO2 treatment? - might author provide some theory?

Answer: We have already shown that TiO2 NPs could associate with receptors, leading to receptor aggregation facilitating the recruitment of more signal transduction molecules and HepG2 cell proliferation and growth. We have added this information in introduction section. (page 2, lane 48)

  • The obtained results should be discussed with a previously published study to indicate the significance of the proposed study and novelty

Answer: According to reviewer’s comment, we have added this point in discussion part. (page 6, lanes 168 to 169)

Sincerely,

Akiyoshi Taniguchi PhD

Research Center for Functional Materials,

National Institute for Materials Science

1-1, Namiki, Tsukuba, Ibaraki 305-0044 Japan

Round 2

Reviewer 1 Report

The authors addressed all my comments.